# Controversies in the Prevention and Treatment of *Clostridioides difficile* Infection in Adults: A Narrative Review

**DOI:** 10.3390/microorganisms11020387

**Published:** 2023-02-03

**Authors:** Taryn B. Bainum, Kelly R. Reveles, Ronald G. Hall, Kelli Cornell, Carlos A. Alvarez

**Affiliations:** 1Jerry H. Hodge School of Pharmacy, Texas Tech University Health Sciences Center, Amarillo, TX 79106, USA; 2College of Pharmacy, The University of Texas at Austin, Austin, TX 78712, USA; 3Pharmacotherapy Education and Research Center, University of Texas Health San Antonio, San Antonio, TX 78229, USA; 4Center of Excellence in Real-World Evidence, Texas Tech University Health Sciences Center, Dallas, TX 75235, USA

**Keywords:** *C. difficile*, probiotics, vancomycin, fidaxomicin, fecal microbial transplant, metronidazole, SER-109, RBX2660, recurrence, economics

## Abstract

*Clostridioides difficile* remains a problematic pathogen resulting in significant morbidity and mortality, especially for high-risk groups that include immunocompromised patients. Both the Infectious Diseases Society of America and the Society for Healthcare Epidemiology of America (IDSA/SHEA), as well as the American College of Gastroenterology (ACG) and the European Society of Clinical Microbiology and Infectious Diseases (ESCMID) recently provided guideline updates for *C. difficile* infection (CDI). In this narrative review, the authors reviewed available literature regarding the prevention or treatment of CDI in adults and focused on disagreements between the IDSA/SHEA and ACG guidelines, as well as articles that have been published since the updates. Several options for primary prophylaxis are available, including probiotics and antibiotics (vancomycin, fidaxomicin). The literature supporting fidaxomicin is currently quite limited. While there are more studies evaluating probiotics and vancomycin, the optimal patient populations and regimens for their use have yet to be defined. While the IDSA/SHEA guidelines discourage metronidazole use for mild CDI episodes, evidence exists that it may remain a reasonable option for these patients. Fidaxomicin has an advantage over vancomycin in reducing recurrences, but its use is limited by cost. Despite this, recent studies suggest fidaxomicin’s cost-effectiveness as a first-line therapy, though this is highly dependent on institutional contracts and payment structures. Secondary prophylaxis should focus on non-antimicrobial options to lessen the impact on the microbiome. The oral option of fecal microbiota transplantation (FMT), SER109, and the now FDA-approved RBX2660 represent exciting new options to correct dysbiosis. Bezlotoxumab is another attractive option to prevent recurrences. Further head-to-head studies of newer agents will be needed to guide selection of the optimal therapies for CDI primary and secondary prophylaxis.

## 1. Introduction

*Clostridioides difficile*, formerly known as *Clostridium difficile*, is currently the most common pathogen in healthcare-associated infections and was deemed an urgent threat in the Center for Disease Control and Prevention’s 2019 report on antibiotic resistance threats in the United States [1]. While the incidence of *Clostridioides difficile* infection (CDI) in the healthcare setting is declining, due to improvements in infection control, antimicrobial stewardship, and diagnostic stewardship, it remains a significant source of morbidity and mortality [1,2,3]. Since the CDC’s initial report in 2013 to its subsequent 2019 report, the number of estimated cases of CDI requiring hospitalization or occurring in hospitalized patients fell from 250,000 to 223,900, while the number of deaths fell from 14,000 to 12,800 [1]. However, recurrence remains a significant obstacle in the treatment of CDI. Rates of recurrence have been estimated to be 20–30% within eight weeks of therapy completion, and the rate of recurrence greatly increases after two or more episodes [4]. Recurrent CDI (rCDI) has also been associated with an increased risk of mortality [5]. European surveillance data from 2016–2017 indicates that almost 60% of CDI cases had contact with healthcare in the 3 months before their hospital admission. In addition, 6% of cases were reported to be recurrent, and recurrent cases were almost twice as likely to have a complicated course of infection [6]. Global rates of healthcare-associated CDI have been estimated at 2.24 per 1000 admissions per year and 3.54 per 10,000 patient days [7].

There were some notable changes for the treatment of initial CDI episodes in the 2017 Infectious Diseases Society of America and the Society for Healthcare Epidemiology of America (IDSA/SHEA) CDI treatment guidelines, as well as in the focused update in 2021, which includes: (1) the removal of metronidazole as a preferred option for mild CDI in 2017 and (2) the addition of fidaxomicin in the 2017 guideline and its preferred status over vancomycin in the 2021 update [8,9]. The conclusions reached by the Infectious Diseases Society of America and the Society for Healthcare Epidemiology of America were developed by only evaluating evidence from randomized controlled trials to conduct their own meta-analyses. However, many controversies for the management of this disease still exist, as are noted by the different conclusions reached by the American College of Gastroenterology (ACG), who utilized real-world data and cost considerations in addition to evaluating data from randomized, controlled trials [10]. The European Society of Clinical Microbiology and Infectious Diseases (ESCMID) focused its recommendations on outcomes more than cost analyses [11]. Summaries of treatment and prophylactic recommendations from each major CDI guideline and this review are presented in Table 1 and Table 2. 

Access to care issues have also been noted, given the high costs for many of the newer therapies being associated with decreased rates of recurrence. This narrative review evaluates the available literature regarding these recommendations to determine the optimal management of CDI in adults based on our assessment of the current evidence. For this narrative review, the authors utilized the PubMed database to search for *Clostridium difficile* or *Clostridioides difficile* when combined with prevention or treatment. References of selected articles were also screened to evaluate other potential articles for inclusion.

## 2. Primary Prevention

### 2.1. Probiotics

Probiotics have been investigated as a method for preventing CDI in patients receiving antibiotic therapy. One theory behind probiotic efficacy is related to the restoration of the gut microbiome that may be disrupted by antibiotic therapy. Other proposed mechanisms of action include protection against pathogens through competition for resources, maintenance of the epithelial barrier of the gut, the production of compounds that inhibit C. difficile growth, and immunomodulation [12]. While the most recent iteration of the IDSA/SHEA CDI guidelines decline to make a recommendation on probiotic use due to insufficient evidence, the ACG guidelines recommend against the use of probiotics for primary or secondary prevention [9,10], and the ESCMID guidelines recommend against probiotics for primary prevention (ESCMID). This recommendation was made because the majority of evidence on the topic is from meta-analyses pooling data from small trials utilizing different strains of probiotics and study methodologies. The majority of trials also evaluated the incidence of CDI as a secondary endpoint and were underpowered. However, interest in probiotics as a means of primary prevention persists due to the relatively low risk of adverse effects and the potential benefit.

Of note, a guideline published by the American Gastroenterology Association (AGA) recommended specific strains and strain combinations for the prevention of CDI, though it was a conditional recommendation based on low quality evidence [13]. The strains endorsed by this guideline were *S. boulardii*, the combination of *L. acidophilus* and *L. casei* LBC80R, the combination of *L. acidophilus*, *L. delbrueckii* subspecies *bulgaricus,* and *B. bifidum*, and the combination of *L. acidophilus*, *L. delbrueckii* subspecies *bulgaricus*, *B. bifidum*, and *S. salivarius* subspecies *thermophilus*. This recommendation was based on a technical review by Preidis and colleagues that analyzed the certainty of evidence from a 2017 Cochrane review [14]. 

The potential benefits of probiotics have been demonstrated in various meta-analyses and retrospective studies. A publication by Maziade and colleagues described 10 years worth of data on over 44,000 inpatients given a probiotic containing *Lactobacillus acidophilus*, *Lacticaseibacillus casei*, and *Lacticaseibacillus rhamnosus* prophylactically within 12 h of each antibiotic course [15]. The CDI rate declined from 18 cases per 100,000 patient-days to 2.3 cases per 100,000 patient-days, and rates of CDI were lower than other comparable Canadian hospitals. 

A meta-analysis that included 21 randomized, controlled trials assessing probiotics for the prevention of CDI found that four of five probiotic types were significantly effective for primary prevention [16]. These included *S. boulardii*, *L. casei*, a mixture of *L. acidophilus* and *B. bifidum*, and a mixture of *L. acidophilus*, *L. casei*, and *L. rhamnosus*. Results for *L. rhamnosus* alone did not show statistical significance for primary prevention. 

The 2017 systematic review and meta-analysis used in the technical review included 31 randomized, controlled trials on probiotic use, which found a CDI incidence of 1.5% in the probiotic group compared to 4% in the control group (RR 0.40, 95% CI 0.30–0.52) [17]. Of note, subgroup analyses revealed that trials enrolling patients with a baseline CDI risk of 0–5% did not show significant differences in efficacy. Trials enrolling patients with a baseline CDI risk (control event rate) of >5% did show significant reduction in CDI rates with probiotics (RR 0.30, 95% CI 0.21–0.42).

Another meta-analysis included 19 studies and over 6000 subjects [18]. The incidence of CDI was lower in the probiotic cohort when compared to the control cohort (1.6% vs. 3.9%, *p* < 0.001). This analysis also indicated that the risk reduction of CDI was greater when probiotics were given closer to the first antibiotic dose. Probiotics given within 2 days of antibiotic initiation resulted in a greater CDI reduction than those given later in the antibiotic course.

Not all studies demonstrated benefit for the primary prevention of CDI. A recent study assessed CDI risk between propensity-matched patients who received probiotics and those who did not receive probiotics and found no difference (OR 1.46, 95% CI 0.87–2.45) [19]. This study only utilized one formulation of probiotic, which included *L. acidophilus*, *L. casei*, and *L. rhamnosus*. In order to avoid serious adverse events, patients were not given the probiotic if they were deemed immunocompromised or critically ill. This could have lessened potential benefits of the probiotic, because those risk factors also increase the risk of CDI. In addition, only 17% of patients in the post-intervention group actually received probiotics, which made it difficult to ascertain the true effect of probiotics in this study.

While the evidence for probiotics as a primary prevention measure remains sparse and conflicting, probiotics are generally considered safe in most populations. Commonly reported adverse effects typically include GI symptoms [17]. Serious adverse effects, such as bacteremia after probiotic administrations, have been described primarily in case reports and are thought to be more of a concern for immunocompromised patients. 

A meta-analysis reported that 32 of the 39 studies included reported data on adverse events [17]. Ten of those studies reported no adverse effects in the treatment or control group, and seven studies reported serious adverse events, none of which were attributable to probiotics. The total incidence of adverse effects in the probiotic group was 14% compared to 17% in the control group (RR 0.83, 95% CI 0.71–0.97). The most common adverse effects reported in both groups were nausea, abdominal cramping, fever, soft stools, flatulence, and taste disturbances.

Another analysis specifically focused on the safety of probiotics when used to prevent or treat disease [20]. This analysis included a total of 622 studies, 235 of which offered nonspecific statements that probiotics were “well tolerated”. Randomized controlled trials primarily reported GI symptoms and showed no significant increase in the overall number of adverse events (RR 1.00, 95% CI 0.93–1.07) or serious adverse events (RR 1.06, 95% CI 0.97–1.16) when short-term probiotics were used. Fungemia and bacteremia potentially associated with probiotic use were reported in some case studies. However, case studies also suggested that immunocompromised patients are the most likely to experience probiotic-related adverse events. The analysis did note that adverse effects were poorly documented in many of the included studies.

It is worth noting that there are several limitations in probiotic use. In addition to the inconsistent strains studied, probiotics are classified as dietary supplements. This means they are not subject to the U.S. Food and Drug Administration (FDA) approval process or good manufacturing practices. Labels for probiotics are not standardized, and, therefore, the exact strains and counts of probiotics may not be reported or readily available.

Though probiotics are not endorsed by guidelines at this time, they remain a potential option to help prevent CDI in immunocompetent populations. Probiotics will likely be of most benefit in institutions or patients with high baseline rates of CDI. More randomized, controlled trials are needed to elucidate definitive benefit and to ascertain which probiotic strains are of most benefit. 

### 2.2. Antimicrobial Prophylaxis

The concept of antimicrobial prophylaxis for other infectious pathogens has been long-standing, particularly for immunocompromised patients. More recently, investigators have begun to evaluate the efficacy and safety of CDI prophylaxis, due to the detrimental effects of CDI. The primary populations who have been evaluated to date are patients who are at an increased risk, such as immunocompromised patients and those undergoing systemic antibiotic therapy. This approach involves the administration of an effective agent against *C. difficile* for a period of time while a risk factor exists in order to prevent a CDI episode. Limited data are currently available on this topic. 

Primary prophylaxis data with fidaxomicin is currently limited to a single randomized, placebo-controlled study of 600 patients undergoing hematopoietic stem-cell transplantation [21]. Fidaxomicin prophylaxis resulted in a decreased rate of confirmed CDI when compared to a placebo at both 30 days (4 vs. 11%, *p* = 0.0014) and at 60 days (6 vs. 11%, *p* = 0.0117). However, the study failed to show a difference in its primary endpoint of incidence of CDI from the first dose of the study drug through 30 days after the last dose of the study drug (29 vs. 31%, *p* = 0.28), since the administration of a CDI-effective medication or missing data were counted as failures.

A meta-analysis of three retrospective cohort studies and one randomized, open-label study evaluated the effectiveness of vancomycin as a prophylaxis [22]. Two of the studies consisted of patients receiving transplants, and the other two included older adults who were receiving systemic antibiotics. The authors found lower rates of CDI in the primary prophylaxis analysis using a random effects model (OR 0.02, 0.00–0.18). The raw event rates were 0.3% for the vancomycin group (1/293) vs. 8.2% for the control group (68/232). The lowest rate of CDI in the control group was 6% in a cohort of patients receiving lung transplantation [23]. The highest rate of CDI in the control group was 20% for patients receiving allogeneic hematopoietic cell transplantation [24]. This study also had the longest duration of vancomycin prophylaxis (29 days), but 75% of the patients in the control arm who developed CDI did so by day 12.

A prior editorial encouraged further research in this area [25]. A head-to-head comparison of vancomycin and fidaxomicin was suggested in terms of antimicrobial prophylaxis. This is needed to determine if one agent is more effective and to determine potential differences in the antimicrobial resistance arising due to prophylaxis. Fidaxomicin’s more narrow spectrum of activity and decreased potential of damage to the commensal gut microbiome when compared to vancomycin may also result in fidaxomicin being a better option for prophylaxis. Other means to decrease the development of antimicrobial resistance include the future possibility of non-antibiotic means of prophylaxis. These include options such as non-toxigenic *C. difficile* administration, tolevamer, and bezlotoxumab. Regardless of the approach taken, the baseline rates of CDI and the real-world impact of primary prophylaxis will need to be assessed from an economic perspective to determine their viability as a standard practice. For now, primary prophylaxis has been consistently shown to be effective in populations with a sufficiently high baseline CDI rate and, therefore, should be considered in high-risk patients.

### 2.3. Proton Pump Inhibitor Discontinuation

Numerous studies have examined the possible association between proton pump inhibitors (PPIs) and CDI and yielded conflicting data. PPIs are common medications in both inpatient and outpatient settings and are thought to contribute to the development of CDI by inhibiting gastric acid production and allowing the proliferation of spores and their conversion to a vegetative form of *C. difficile* [26]. There is also evidence to suggest that even a short course of PPIs can significantly alter gut microbiota and result in lower concentrations of *Lachnospiraceae*, *Erysipelotrichaceae*, and *Bifidobacteriaceae* as well as higher concentrations of *Streptococcaceae* [27]. These changes in gut microbiome mimic ones seen in patients with CDI. Both the IDSA and ACG CDI guidelines state there is insufficient evidence to recommend the discontinuation of PPIs as a prevention measure, but they do recommend that PPI therapy should only be continued when a legitimate indication is present [8,10]. The ESCMID guidelines recommend that PPI therapy should be reviewed and do include PPI use as a prognostic factor that can signal an increased risk of CDI [11].

One meta-analysis published in 2017 included 56 studies examining CDI incidence with PPI use [26]. The pooled analysis showed a significant increase in CDI when PPI users were compared to non-PPI users (OR 1.99, 95% CI 1.73–2.30). Subgroup analyses were performed on cohort and case-control studies, studies with adjusted and unadjusted odds ratios, single-center and multi-center studies, inpatient and outpatient populations, various geographic regions, and high-quality studies. All of these subgroups yielded statistically significant results that indicated the increased risk of CDI with PPI use. 

Another systematic review and meta-analysis focused on 12 studies that reported hospital-acquired CDI in patients receiving stress ulcer prophylaxes with either PPIs or histamine-2 receptor antagonists (H2RAs) [28]. In this analysis, PPIs were associated with an increased incidence of CDI (OR 1.386, 95% CI 1.152–1.668). This association remained significant when various subgroups were analyzed, including acid suppression for stress ulcer prophylaxis versus acid suppression for unspecified purpose, intensive care unit patients versus non-intensive care patients, and types of study designs. 

A recent cohort study reviewed data for patients in Denmark and identified 3583 episodes of community-acquired CDI [29]. Of those cases, 964 occurred with current PPI use, 324 occurred up to 6 months after PPI discontinuation, 123 occurred 6–12 months after discontinuation, and 2172 occurred without PPI use. The study indicated that the adjusted incidence rate ratio (IRR) for CDI with PPI use compared to non-use was 2.03 (95% CI 1.74–2.36). The risk persisted for the groups that were within 6 months of discontinuation (IRR 1.54, 95% CI 1.31–1.80) and for the group within 6–12 months of discontinuation (IRR 1.24, 95% CI 1.00–1.53). The conclusion was that PPI use was associated with increased CDI risk, even up to a year after the discontinuation of therapy.

In 2021, a review was published that summarized eight systematic reviews and meta-analyses regarding this topic [30]. All eight showed statistically significant associations between PPI use and the incidence of CDI. The review classified the associations found as minimal (OR 1.1 to 1.49), moderate (OR 1.5 to 1.9), and high (OR > 2.0). Risks found in the meta-analyses ranged from minimal to high, but the majority fell in the moderate risk category.

One prospective, randomized trial referenced by the ACG CDI guidelines that evaluated the safety of PPIs in patients with stable cardiovascular disease noted a statistically significant increase in the incidence of enteric infections in PPI users compared to the control group (1.4% vs. 1.0%, *p* = 0.04) [31]. The incidence of CDI was low overall, and the numerically higher incidence in PPI users was not statistically significant (0.1% [9/8791] vs. <0.1% [4/8807], *p* = 0.18). However, this study was in an outpatient population who may have had a low baseline risk for CDI. In addition, other factors, such as antibiotic courses that could have impacted CDI risk, were not collected. 

An additional meta-analysis evaluated 39 studies to determine the risk of CDI with PPI use [32]. A pooled analysis showed a significant association with PPI use and CDI risk (OR 1.74, 95% CI 4.17–2.85, I2 = 85%) when compared to non-PPI users. This analysis also indicated an increased risk of rCDI in PPI users (OR 2.51, 95% CI 1.16–5.44, I2 = 78%). This study performed an adjusted indirect analysis to compare CDI rates among histamine 2 receptor antagonists (H2RAs) and PPIs and found H2RAs to have a lower risk (OR 0.71, 95% CI 0.53–0.97). 

Though causality has not been proven, and many potential confounding factors exist, the literature does seem to suggest that acid suppression therapy with PPIs is associated with an increased risk of developing CDI. We agree with the American College of Gastroenterology CDI guidelines that PPIs are useful therapies that should be used only when an indication truly exists. However, we feel the need for PPI vs H2-antagonist therapy should be weighed against the risk of adverse effects such as CDI, especially in patients with multiple risk factors for CDI.

## 3. Treatment of an Initial CDI Episode

Three antibiotics are endorsed by guidelines in the treatment of CDI–metronidazole, oral vancomycin, and fidaxomicin. Metronidazole is a nitroimidazole antibiotic that passively diffuses into organisms and is reduced, thus resulting in free radical formation. The reduced form of metronidazole and the free radicals interact with bacterial DNA to lead to the inhibition of DNA synthesis, the degradation of DNA, and bacterial death [33]. Because metronidazole is active against most anaerobic organisms, it has the potential to disrupt the gut microbiome significantly. Vancomycin, a glycopeptide antibiotic, inhibits cell-wall biosynthesis and alters bacterial cell membrane permeability and RNA synthesis (package insert) [34]. This agent also has the potential to significantly disrupt gut flora, as it is active against many bacteria known to inhabit the GI tract. Fidaxomicin is a macrolide antibiotic that binds to RNA polymerases and inhibits RNA synthesis [35]. Fidaxomicin has a narrow spectrum of activity, thus minimizing its effect on gut microbiota and, therefore, preventing CDI recurrence. 

### 3.1. CDI Severity Definitions

Both the 2017 IDSA and 2021 ACG guidelines make the distinction between non-severe and severe CDI based on the patient’s white blood count and serum creatinine [8,10]. The ESCMID guidelines also utilize these criteria to define severe infection with the addition of fever and certain indications of inflammation from imaging [11]. Severe CDI is noted as a white blood cell count of ≥15,000 cells/mL or a serum creatinine level >1.5 mg/dL. None of the guidelines define moderate CDI. The literature in the treatment sections below may vary slightly in their definitions of CDI severity, but these are currently the most commonly accepted severity definitions.

### 3.2. Is There a Role for Metronidazole in Non-Severe CDI?

Metronidazole had been the first-line treatment for mild CDI for almost three decades based on two randomized, controlled, unblinded trials published in 1983 and 1996 [36,37]. The first trial compared 10-day courses of metronidazole (250 mg PO QID, *n* = 42) and vancomycin (500 mg PO QID, *n* = 52) [36]. Treatment failure rate was numerically higher for metronidazole and vancomycin (4.8% vs. 0%, *p* = 0.20), but did not reach statistical significance. Relapse rates were numerically lower for metronidazole (4.8% vs. 11.5%, *p* = 0.17), but also failed to reach statistical significance. The study also cited a lower cost associated with metronidazole versus vancomycin. The second trial compared 10-day courses of fusidic acid, metronidazole (500 mg PO TID, *n* = 31), teicoplanin, and vancomycin (500 mg PO TID, *n* = 31) [37]. Clinical cure rates (94%) and recurrence rates (16%) were the same for metronidazole and vancomycin. The consensus, based on these two small randomized, controlled trials, was that metronidazole was as effective as and less costly than vancomycin for initial disease and that vancomycin should be reserved for cases of metronidazole failures or intolerance. These studies did not stratify patients by disease severity, and they also took place prior to metronidazole-related minimum inhibitory concentration (MIC) elevations in some *C. difficile* strains [38].

The first blinded randomized, controlled trial of metronidazole for CDI that was stratified based on disease severity was published in 2007 [39]. The trial found significantly improved cure rates with vancomycin overall (97% vs. 84%, *p* = 0.006). Cure rates were statistically different between vancomycin and metronidazole for severe disease (97% vs. 76%, *p* = 0.02), but not for mild disease (98% vs. 90%, respectively, *p* = 0.36). Vancomycin also had numerically lower relapse rates (7% vs. 14%, *p* = 0.27) that did not reach statistical significance. This was in part due to only 14 patients experiencing a recurrence. The results of this trial suggested that metronidazole may be an acceptable option for initial mild disease.

Two randomized, double-dummy, active-controlled, parallel design studies compared tolevamer, metronidazole, and vancomycin for the treatment of CDI [40]. Clinical success rates between the two studies showed statistically significant results in favor of vancomycin (81% vs. 73%, *p* = 0.02). When infection was stratified by severity, the differences were no longer statistically significant; however, vancomycin had numerically higher clinical success for all groups (mild: 83% vs. 79%, *p* = 0.54, moderate: 82% vs, 74%, *p* = 0.14, severe: 79% vs. 66%, *p* = 0.06). The absolute difference in clinical success rates increased as disease severity increased. Similar results were observed when evaluating recurrence. These results reiterate the possibility that metronidazole is a reasonable option for mild disease.

A retrospective propensity-matched cohort study using Veterans Affairs data compared 30-day mortality between vancomycin and metronidazole in cases of CDI [41]. Thirty-day mortality was lower for vancomycin when evaluating all CDI cases (9% vs. 11%, *p* = 0.01). There was no difference in mortality observed when only evaluating mild or moderate disease (vancomycin: 6%, vs. metronidazole: 7%, *p* = 0.22). Vancomycin was associated with lower mortality rates (15% vs. 20%, *p* = 0.01) in patients with severe disease. Vancomycin was still associated with lower risk of death for severe disease in a multivariable Poisson regression model (adjusted RR, 0.79; 95% CI, 0.65–0.97). This difference was not maintained for patients with mild to moderate disease. Another multivariable Poisson regression model from this study suggested that vancomycin did not significantly reduce CDI recurrence for any disease severity. The results of this study support the use of vancomycin in severe disease, but again fail to suggest that metronidazole is an inferior option for mild disease. 

Another retrospective cohort study assessed risk factors for 30-day all-cause mortality in 924 patients following CDI at 42 facilities in Japan [42]. The results of a logistic regression analysis showed a significant reduction in mortality when vancomycin monotherapy (*n* = 433) was administered when compared to cases where no anti-CDI drugs were administered (OR 0.43; 95% CI, 0.25–0.75). Metronidazole monotherapy (*n* = 237) was not found to significantly alter mortality when compared with no anti-CDI drug (OR 0.85; 95% CI, 0.48–1.51). However, the severity of disease was not delineated in this study.

Zhang and colleagues evaluated vancomycin vs metronidazole in both severe and non-severe patients with CDI [43]. The treatment choice did not meet the pre-specified threshold for inclusion in the multivariable analysis of all-cause 30-day mortality or recurrence in the overall cohort. The authors noted that vancomycin was associated with a decreased risk of new infection (9–26 weeks after CDI treatment initiation) in a subgroup multivariable logistic regression analysis of patients with non-severe episodes (OR 0.11, 95%CI 0.02–0.86). This result does not agree with prior findings and may have been influenced by not including the follow-up time of each patient in the analysis by using a Cox proportional hazards model. The higher vancomycin 30-day all-cause mortality rate may have decreased the follow-up time in this subgroup. In addition, data regarding the length of stay were not shown to be able to determine if any differences existed between the groups, given that outpatient data were not collected for the cohort.

There have historically been few reliable data available regarding the impact of metronidazole MICs on clinical outcomes. Gonzalez-Luna and colleagues evaluated the impact of metronidazole MICs on clinical failure, which was defined as the presence of CDI-specific symptoms on day 6 of treatment or later, a change in CDI therapy due to a lack of patient response before day 7, and/or CDI-contributable mortality within the first 7 days of treatment, in 356 patients with CDI [44]. A classification and regression tree univariable analysis revealed that a metronidazole MIC of 1 mg/L or greater was associated with clinical failure. This was confirmed in the multivariable analysis of the 255 patients receiving metronidazole (OR 2.27, 95% CI 1.18–4.34). However, a similar trend was observed in the 101 patients not receiving metronidazole (OR 3.15, 95% CI 0.83–11.94), which suggested that other immunomodulatory factors may be responsible for both the metronidazole MIC increase and the rise in failure rates. The application of these findings may also be limited by clinical laboratories not routinely determining metronidazole MICs and by the use of a novel method for metronidazole MIC determination being used that may not have been routinely adopted yet by clinical laboratories.

The overall clinical failure rate of metronidazole for CDI has increased over time. However, these higher failure rates may be due to an increased frequency of severe disease. The above studies suggest that, while overall clinical failure rates may be increased with metronidazole compared to vancomycin, this may not hold true for mild disease. This highlights the importance of determining the severity of disease when selecting a treatment regimen for CDI. 

### 3.3. Should Vancomycin Be Relegated to a Second-Tier Option for the Treatment of an Initial CDI Episode?

#### 3.3.1. Clinical Data in Non-Severe Infections

Fidaxomicin was approved by the US FDA in 2011 shortly after the 2010 CDI treatment guidelines update [45]. The recommendation for fidaxomicin as a first line agent in the 2017 guidelines is in part supported by its narrow spectrum of activity, limited systemic absorption, and ability to inhibit sporulation [46]. Both phase 3 randomized, controlled trials supporting this change were non-inferiority studies [47,48]. The two arms were 200 mg of fidaxomicin administered twice daily and 125 mg of standard oral vancomycin administered four times daily for 10 days and stratified by primary infection or first recurrence. Collectively, these international trials included 1164 inpatients and outpatients.

Louie et al. reported that clinical cure rates for fidaxomicin met non-inferiority in both the modified intention-to-treat (mITT) analysis (88% fidaxomicin vs. 86% vancomycin) and per-protocol (PP) analysis (92% vs. 90%) [47]. Fidaxomicin also produced similar results to vancomycin when evaluating secondary outcomes. Recurrence rates at 28 days following the completion of the study treatment were significantly lower in the fidaxomicin group (mITT: 15% vs. 25%, *p* = 0.005); in addition, global cure rates were significantly higher in the fidaxomicin group (mITT: 75% vs. 64%; *p* = 0.006). The majority of patients included were classified as mild or moderate CDI at baseline based on the number of bowel movements per day and white blood cell count. Post-hoc subgroup analysis by disease severity did not reveal any significant differences between groups for clinical cures (*p* values not reported). Post-hoc analysis of the recurrence rates by subgroups revealed overall lower rates with fidaxomicin in patients with mild disease in the mITT group (12% vs. 29%; *p* = 0.02) and PP group (9% vs. 24%, *p* = 0.06). 

Cornley et al. also found non-inferiority was met with fidaxomicin for clinical cures in both the mITT (92% fidaxomicin vs. 91% vancomycin) and PP analyses (88% vs. 87%) [48]. The recurrence rate for vancomycin was significantly higher than for fidaxomicin (mITT: 27% vs. 13%, *p* = 0.0002). Fidaxomicin maintained superior results for sustained response rates (77% vs. 63%, *p* = 0.001). Approximately three-quarters of patients had non-severe disease at baseline based on WBC, SCr, and temperature. Post-hoc subgroup analysis revealed no difference in the primary endpoints of clinical cures in patients with non-severe CDI (92% vs. 92%, *p* = 0.914). The subgroup analysis of recurrence rates by severity showed no evidence of heterogeneity and, therefore, was not analyzed for significance based on study protocol. However, recurrence rates were numerically lower in the fidaxomicin group for non-severe CDI (14% vs. 26%). Lastly, sustained response rates remained consistent, favoring fidaxomicin, when assessed by CDI severity (non-severe: 79% vs. 68%; *p* = 0.02). 

Results from the above studies were combined using fixed-effects meta-analysis [49]. Overall results were consistent, and reproduced the noninferiority of fidaxomicin to vancomycin for clinical cures (*p* < 0.0001). Fidaxomicin had a reduced risk of recurrence (RR 0.54; 95% CI, 0.42–0.71), and “no global cure” (e.g., failure) (RR 0.68; 95% CI, 0.56–0.81). A post-hoc, exploratory time-to-event, intention-to-treat analysis was also conducted and showed a 40% decrease (95% CI, 26–51%; *p* < 0.0001) in persistent diarrhea, recurrence, and death through day 40. Subgroup analysis results by severity did not remarkably vary from the individual studies. The outcomes of persistent diarrhea, recurrence, or death through day 40 were significantly lower in the fidaxomicin group for both mild disease (RR 0.46; 95% CI 0.30–0.70) and moderate disease (RR 0.66; 95% CI 0.47–0.92).

A third phase 3, double-blind, multicenter, non-inferiority study comparing standard vancomycin to fidaxomicin conducted in hospitalized patients in Japan had slightly differing results [50]. The majority of patients (76%) included had non-severe disease. Based on a margin of 10%, non-inferiority was not achieved in the population who received at least 1 dose of fidaxomicin for the primary endpoint of the CDI global cure rate, which was defined as the proportion of patients cured at the end of treatment without recurrence during a 28-day follow up (67% fidaxomicin vs. 66% vancomycin; 1.2% difference; 95% CI, −11.3–13.7). Global cure was, however, higher for fidaxomicin in the per-protocol analysis (74% vs. 70%) and post-hoc analysis of the patients who received at least 3 days of treatment (72% vs. 67%). Power was not met in the fidaxomicin group by one subject, but this small difference is not likely to change our interpretation of the results. Recurrence rates were lower for fidaxomicin at 28-days in the population who received at least 1 dose of fidaxomicin (20% vs. 25%), with similar results for secondary analysis populations.

There is evidence that extending 20 fidaxomicin doses over a longer period of time after initial daily dosing may allow for the recovery of gut microbiota and allow fidaxomicin to persist at concentrations that are adequate to inhibit *C. difficile* [51]. A randomized, controlled, open-label, multicenter, European superiority trial (EXTEND) studied hospitalized CDI patients aged 60 years and older [48]. The trial compared extended-pulsed fidaxomicin dosing (200 mg twice daily on days 1–5, then once daily on alternate days on days 7–25) to standard oral vancomycin, which was stratified by CDI severity, presence of cancer, age, and CDI recurrence. Seventy percent of patients in the fidaxomicin group sustained clinical cures for 30 days after treatment completion compared to 59% in the vancomycin group (OR 1.62; 95% CI, 1.04–2.54). Significant results were consistent across the PP population and extended through day 90 (66% vs. 51%, *p* = 0.007). Additional subgroup analyses using the Cochran–Mantel–Haenszel test, adjusted for baseline stratification, resulted in no difference for sustained cures at 30 days in patients with non-severe CDI (75% with fidaxomicin vs. 64% with vancomycin, *p* = 0.068). The recurrence rate for the fidaxomicin arm in this trial (6%) is lower than recurrence rates reported in previous studies [43,45].

A single-center, retrospective cohort study in the Czech Republic published in 2021 evaluated four CDI treatment groups (oral metronidazole, vancomycin, combination of intravenous metronidazole and vancomycin, and fidaxomicin) [52]. Eighty-two percent of patients had non-severe disease, based on an ATLAS score of 0–5. Fidaxomicin had significantly higher sustained clinical cure rates when compared to all other treatment arms, including vancomycin monotherapy (79% vs. 44%; *p* = 0.0001), for the primary outcome in the overall study population. Similarly, fidaxomicin was associated with lower recurrence rates across the board in the overall study population, including vancomycin (14% vs. 46%; *p* = 0.0003). Subgroup analysis showed continued significant differences, favoring fidaxomicin over vancomycin, in non-severe CDI for sustained clinical response (83% vs. 47%; *p* = 0.0079) and recurrence (13% vs. 46%; *p* = 0.0006). 

Though the first phase-3 trials were essential for the evolution of updated CDI treatment guidelines, it is important to take into consideration that the outcomes according to severity were only assessed via post-hoc subgroup analyses. Additionally, there is a lack of universal severity definitions among both practice guidelines and clinical trials, as Chopra et al. describes [53], but a common theme among severe CDI groups (excluding use of the ATLAS score) is leukocytosis defined as >15,000 cells/mm^3^. Repeatedly, the literature demonstrates similar clinical cure rates between fidaxomicin and vancomycin for CDI treatment regardless of disease severity, including consistency from several meta analyses [54,55,56]. While results for clinical cures are impressive in regards to the novel macrolide, the literature suggests that the niche for fidaxomicin lies within preventing disease recurrence.

#### 3.3.2. Clinical Data in Severe Infections

Louie et al. showed lower recurrence rates with fidaxomicin for severe CDI in the mITT population (13% vs. 27%; *p* = 0.02) based on post-hoc analyses. [47]. Post-hoc subgroup analyses showed no significant difference for the primary endpoint of clinical cure (mITT: 76% vs. 71%, *p* = 0.473) but favored fidaxomicin for recurrence (8% vs. 33%, *p* = NA) in patients with severe disease [48]. 

A retrospective propensity-score matched analysis in a Veterans Affairs cohort compared standard oral vancomycin to fidaxomicin in severe CDI patients, who were classified in accordance with the 2010 SHEA/IDSA CDI guidelines [57]. No difference was found between fidaxomicin and vancomycin for combined 90-day recurrence or clinical failure (32% vs. 26%, *p* = 0.07). Similarly, no significant difference resulted between treatment groups for secondary outcomes of 30, 90, and 180-day mortality. 90-day recurrence rates were identical between groups (24%, *p* = 1.0); however, clinical failure rates were higher in the fidaxomicin group, which was significant (9% vs. 1%, *p* < 0.001). It is important to note that the definition of clinical failure was a change in therapy (addition of metronidazole or conversion to vancomycin or fidaxomicin) after 3 days of initiation and that the rationale behind therapy changes were not delineated.

Multivariate analysis of the EXTEND trial in the modified full analysis population resulted in a significant difference between severe vs. non-severe CDI, and showed severe CDI was less likely to achieve sustained clinical cures at 30 days following treatment completion (OR 0.57; 95% CI, 0.36–0.91). These results were not retained in the per-protocol data set (OR 1.33; 95% CI, 0.49–3.60). Subgroup analyses adjusting for baseline stratification were unable to find any significant difference for sustained cure at 30 days for severe CDI (62% fidaxomicin vs. 51% vancomycin, *p* = 0.235) [58]. 

Subgroup analysis for severe CDI in the single-center Czech Republic study varied from results in the non-severe disease group [52]. Statistically significant differences were not found between fidaxomicin (*n* = 9) and vancomycin monotherapy (*n* = 16), respectively, for sustained clinical response (56% vs. 31%; *p* = 0.24) or recurrence (17% vs. 44%; *p* = 0.28). Less than 20% of subjects in the study had severe disease (*n* = 49). A small number of study subjects is a limitation that should be taken into consideration, as there were numerically large differences favoring fidaxomicin. 

The representation of severe CDI is largely limited in current studies, with the majority of CDI patients categorized as mild or moderate, and CDI severity is commonly assessed via small subgroup analyses. Therefore, the literature is less clear when making conclusions regarding fidaxomicin utility in severe CDI.

### 3.4. Economic Considerations: Is the Fidaxomicin Clinical Benefit Worth the Increased Drug Cost?

There are numerous economic evaluations of fidaxomicin for CDI. This summary will focus on studies that primarily evaluated the first CDI episode. Reveles, et al. performed a pharmacoeconomic analysis to determine the cost effectiveness of fidaxomicin versus vancomycin as a first-line treatment for CDI [59]. This model assumed a drug acquisition cost of USD 235 per day for fidaxomicin and USD 20 to USD 40 per day for vancomycin. This analysis found the overall costs to be similar for both options when accounting for initial hospitalization, drug acquisition cost, and rehospitalizations for recurrences. The costs differed by less than USD 300 per patient when analyzing various subgroups for the potential optimization of fidaxomicin use and resulted in cost savings for patients with cancer and those with concomitant antibiotic use.

Another cost-effectiveness analysis was performed to compare fidaxomicin, bezlotoxumab plus vancomycin, extended-pulsed fidaxomicin, and standard oral vancomycin for the initial treatment of CDI [60]. This model found standard fidaxomicin to be the most cost-effective treatment when considering cost and quality-adjusted life-years. Extended-pulsed fidaxomicin and bezlotoxumab–vancomycin were found to be more cost effective than standard vancomycin therapy. The results of this analysis may not be applicable to a patient population with high rates of BI/NAP1/027 strains, as data suggests fidaxomicin’s benefit of lower recurrence rates is limited with this strain.

A cost equivalency study conducted by Patel and colleagues examined the efficacy of fidaxomicin versus vancomycin by assessing all randomized controlled trials up until the time of the 2021 IDSA and SHEA guideline updates [61]. The study also conducted a systematic review of the cost associated with CDI recurrence in order to determine if the cost savings from prevention of recurrence offset drug acquisition cost. The results indicated that it would cost approximately $43,904 to prevent one recurrence by using fidaxomicin as a first-line agent over vancomycin with current drug acquisition costs, and their conclusion was that increased expenditure on fidaxomicin as a first-line treatment would not be offset by the prevention of recurrence. Economic analyses of only the first CDI recurrence likely significantly reduces the cost-effectiveness of fidaxomicin, given that patients who experience one recurrence are at significant risk of subsequent recurrences.

While the overall cost of CDI infections on the healthcare system may be lower with the use of fidaxomicin over vancomycin, these reductions are highly dependent on the increased costs associated with CDI recurrence. Unfortunately, many healthcare systems do not have a direct tie between these two cost centers, and the cost savings associated with decreased recurrences does not impact the decision as to whether to utilize fidaxomicin for an initial episode due to the drug acquisition cost. Another demotivator for fidaxomicin use is the lack of a readmission penalty by the Centers for Medicare and Medicaid Services for CDI, unlike for other conditions, such as myocardial infarctions or community-acquired pneumonia. The lack of a penalty actually provides the hospital or clinic another payment for the recurrence, which de-emphasizes the importance of preventing these negative clinical outcomes. Closed-loop systems, such as the VA or Kaisar, may have a better likelihood of being able to convince system administrators of the increased upfront costs to prevent downstream recurrences and their associated costs.

#### Clinical Data for FMT for Initial CDI Episode

Though most data for FMT is in patients with multiple recurrences, there is a small amount of data for this therapy in initial CDI. 

A small retrospective pre-/post-FMT study described 59 patients who were treated for initial CDI with a 10–14 day course of antibiotics followed by 2 FMT infusions [62]. Of the 54 patients who completed a stool test 4–8 weeks post treatment, 98% were negative for *C. difficile*. Twenty-four patients were followed for 6 months and showed significant improvements in abdominal pain, diarrhea, and blood in the stool from the baseline. No adverse events were reported in the study.

Another retrospective study examined FMT based on an institutional protocol where patients were offered FMT if they had 3 or more recurrences, 2 or more hospitalizations with severe CDI, or if they were experiencing their first episode of CDI that was complicated [63]. Of the 35 patients in the study, 4 underwent FMT for a primary CDI episode. Two of those patients were cured after FMT and the other two worsened and required colectomies (1 death and 1 cure after colectomy). It is difficult to draw conclusions from this study on the efficacy of FMT for a primary episode of CDI, as there were very few patients treated in this manner, and other studies have shown that complicated CDI is associated with a higher rate of FMT failure [64]. 

Another retrospective study examined 96 patients who underwent FMT following a 10-day antibiotic course for CDI [65]. Twenty-five of those patients received FMT for a primary occurrence of CDI (19 had severe CDI and 6 had non-severe). The overall success rate for FMT in a primary CDI episode was 92% (95% in severe cases and 83% in non-severe cases). Though this study had a small number of patients receiving FMT for primary CDI, it adds to the literature supporting FMT for this indication.

The guidelines only endorse FMT for use in recurrent or refractory CDI at this time. However, as FMT becomes more ubiquitous, its role in primary CDI may become clearer.

### 3.5. Biomarkers

The BI/NAP1/027 strain of *C. difficile* has mutations that result in increased toxin production, increased sporulation, and increased uptake of toxins into cells. Because of these changes, there is concern regarding the increased spread of this strain as well as a more severe presentation of the infection [66]. The incidence of this strain was reported at 26% in 2013 and decreased to 17% in 2016 [67]. Conflicting results have been demonstrated for this strain, with some studies showing increased severity of disease, mortality, and recurrence rates and others showing no difference in these outcomes. 

The randomized controlled trial conducted by Louie and colleagues comparing fidaxomicin to vancomycin had a BI/NAP1/027 strain incidence of 36% [47]. Among those with the NAP1 strain, clinical cure rates at the end of therapy did not significantly differ between the fidaxomicin group and the vancomycin group in the modified intention-to-treat population (79% vs. 81%) or the per-protocol population (86% vs. 85%). Recurrence rates also did not significantly differ between fidaxomicin and vancomycin in this population (modified intention-to-treat rate was 27% vs. 21%, *p* = 0.42; and per-protocol rate was 24% vs. 24%, *p* = 0.93). Overall, cure rates for both fidaxomicin and vancomycin were higher in the non-NAP1 strains in this study.

Another study collected isolate information from patients enrolled in two phase 3 clinical trials comparing fidaxomicin to vancomycin for CDI [68]. BI/NAP1/027 accounted for 34% of the isolates identified. Patients infected with the NAP1 strain had significantly lower cure rates compared to patients with non-NAP1 strains (87% vs. 94%, *p* < 0.001). This difference in cure rates held true regardless of treatment with vancomycin (86% in NAP1 vs. 93% in non-NAP1, *p* = 0.02) or fidaxomicin (88% in NAP1 vs. 95% in non-NAP1, *p* = 0.007). 

There is also data to suggest that the combination of eosinopenia and infection with a binary toxin strain of *C. difficile* increases the risk of mortality [69]. A multicenter, retrospective cohort study of 688 patients examined the relationship between infection with binary toxin strains of *C. difficile*, eosinopenia, and inpatient mortality. One hundred and thirty-two patients (19%) were found to have eosinopenia, and one hundred and nine patients had binary toxin strains (16%). The combination of eosinopenia and binary toxin strain was found to be an independent predictor of inpatient mortality (OR 7.8, 95% CI 1.9–33.2). However, only 14 patients had this combination. An analysis was also performed on a VA cohort (*n* = 790) and found that inpatient mortality was significantly increased (OR 6.1, 95% 1.5–23.9) for the 13 patients who had this combination. The combination being observed in only 1.6–2.0% of patients likely limits the general application of these findings, particularly given that most clinical laboratories do not test for CDI strain type.

Though this particular strain of *C. difficile* may put patients at risk for more severe disease or worse outcomes, the treatment remains largely the same as for non-NAP1 strains and, therefore, is not routinely tested for in clinical laboratories. The IDSA guidelines do not recommend a certain agent for the NAP1 strain, nor is there abundant guidance in the literature on which agent is best suited to treat it. We also do not believe that a strain-specific treatment approach is useful at this time.

## 4. Treatment of Recurrent CDI Episode

### What Is the Preferred Treatment Regimen for Recurrent Infections?

Recurrence is a significant obstacle in the treatment of CDI. A systematic review and meta-analysis analyzing 33 studies with 18,530 CDI patients found advanced age, non-CDI antimicrobial therapy during follow up, and the use of a proton pump inhibitor during follow up to be the most common risk factors for rCDI. Renal insufficiency and prior use of a fluoroquinolone were also found to increase risk of recurrence [4]. Because this is a common complication of CDI, it is important to identify those at risk of recurrence as well as understand how best to treat these patients.

The IDSA 2021 focused its update guidelines on the management of CDI to recommend the treatment of first recurrences with either a standard course or extended-pulsed regimen of fidaxomicin (conditional recommendation, low certainty evidence), and vancomycin is listed as an acceptable alternative. Tapered and pulsed vancomycin, vancomycin followed by rifaximin, fidaxomicin, and FMT are all mentioned as treatment options for multiple recurrences of CDI. However, FMT is the only strong recommendation with a moderate quality of evidence [9]. The ESCMID guidelines recommend treatment, in part, based on what therapy was used for the initial episode [11]. Options include fidaxomicin, FMT, or the addition of bezlotoxumab.

The optimal treatment of recurrent CDI episodes is unclear. What is known is that fidaxomicin, whether administered by a standard or pulsed dosing regimen, has a lower recurrence rate than using a standard vancomycin regimen (125 mg PO QID) in randomized, controlled trials. Recurrent CDI subgroup results for these studies echo that fidaxomicin results in lower recurrence rates than a standard vancomycin regimen [48]. The recurrence rate at day 90 for patients with recurrent CDI receiving pulsed fidaxomicin dosing was 13% (4/31), as compared to 32% (10/31) for vancomycin, (no *p*-value provided). Only patients with a first or second recurrence were eligible for inclusion in the original trial. Another post-hoc analysis of patients treated for a first recurrence of CDI within 28 days from two randomized, controlled trials found that recurrence rates were lower with fidaxomicin [70]. While the statistical significance of the result was dependent on whether the modified intention-to-treat (20 vs. 32%, *p* = 0.08) or the per-protocol population (20 vs. 36%, *p* = 0.045) was evaluated, both results are clinically meaningful reductions in recurrence rates.

The role of pulsed or tapered vancomycin regimens is not as clear. The main source of evidence is a post-hoc analysis of two randomized, controlled trials evaluating *Saccharomyces boulardii* vs a placebo [71]. Both arms received a standard antimicrobial for CDI at the time (vancomycin or metronidazole). The mean days of a vancomycin taper or pulse were 19.5 and 20.3, respectively. Lower rates of recurrence within two months were found with the taper (31%, 9/29, *p* = 0.01) or pulse (14%, 1/7, *p* = 0.02) dosing of vancomycin in univariable analyses. However, the methodologic limitations, small numbers of patients receiving tapered or pulse therapy, and the lack of these results being replicated lessen the confidence that can be placed in these findings. 

A retrospective cohort of 100 patients treated with a tapered and pulsed vancomycin regimen following recurrent CDI evaluated a taper of vancomycin to once daily, followed by every-other-day (QOD) dosing, or once daily followed by QOD followed by every-third-day (Q3D) dosing [72]. The total duration of treatment was longer for the Q3D group (86 vs. 60 days, *p* = 0.0004). Cure rates were found to be higher for the Q3D dosing group (81 vs. 61%, *p* = 0.03) in a univariable analysis. Recurrence rates during pulse therapy were lower for the Q3D dosing (8 [5/64] vs. 19% [7/36], no *p*-value provided).

## 5. Secondary Prevention

### 5.1. Probiotics

Data examining the effect of probiotics on the secondary prevention of CDI is sparse and lacks consistency with regard to dosing and strains used. A systematic review and meta-analysis previously discussed assessed four randomized, controlled trials pertaining to probiotics for secondary prevention of CDI [16]. Only *S. boulardii* and *L. rhamnosus* had enough data to assess for secondary prevention. This analysis indicated no benefit of probiotics on rates of recurrent CDI when trials of the data were pooled by similar strain type. All three major CDI guidelines recommend against probiotic use for CDI secondary prevention based on the insufficient evidence of benefit [8,9,10,11]. 

### 5.2. FMT

Available guidelines endorse the use of FMT only in patients with recurrent CDI that is unresponsive to antibiotic therapy [8,9,10,11]. The IDSA guidelines recommends this be after three courses of antibiotic therapy for CDI, though they emphasize this is expert opinion and not necessarily based on data. The ACG guidelines specifically discuss repeated FMT courses in severe or fulminant cases of CDI. The ESCMID guidelines mention FMT as a rescue therapy for severe/fulminant cases of CDI if the patient deteriorates despite antibiotics, particularly if surgery is not an option [11]. FMT is also included as an option after at least 2 recurrences.

A 2017 systematic review and meta-analysis included 37 studies, 7 of which were randomized, controlled trials [73]. The analysis found FMT to be more effective than vancomycin alone at resolving recurrent and refractory CDI, regardless of route of administration (RR 0.23, 95% CI 0.07–0.80). The clinical resolution rate with FMT was 92% across all studies (95% CI 89–94%). Clinical resolution was significantly higher when lower GI delivery was compared to upper GI delivery (95% vs. 88%, *p* = 0.02), but no difference was seen between fresh and frozen FMT (92% vs. 93%, *p* = 0.84). 

Another meta-analysis included 6 studies and found donor FMT to have the highest efficacy when compared to vancomycin (OR 20.02, 95% CI 7.05–70.03) or fidaxomicin (OR 22.01, 95% CI 4.38–109.63) [74]. 

A systematic review and meta-analysis published in 2019 included 13 trials and a total of 610 patients who were treated with single FMT [75]. The overall cure rate was 76% (95% CI 66.4–85.7%). It was noted that cure rates were lower in randomized trials than in open-label studies (68% vs. 82%, *p* < 0.001). Subgroup analyses performed indicated that delivery via colonoscopy had higher cure rates than delivery via enema (87% vs. 66%, *p* < 0.001) and that there was no difference in cure rates between delivery via colonoscopy and oral delivery (87% vs. 81%, *p* = 0.17). 

Efficacy of FMT may increase when multiple doses are given. Several studies have demonstrated an increase in clinical resolution rates when a second dose of FMT was administered in response to clinical failure with the first dose. Clinical resolution rates with one dose ranged from 63–81% and increased to 90–100% with a second dose [76,77,78]. A sequential FMT protocol was assessed and had an overall success rate at 30 days of 91% (100% in severe CDI and 87% in severe/complicated CDI) [79]. In this study, success was achieved with one dose of FMT in 53% of patients and achieved with two doses of FMT in 28% of patients. Vancomycin was administered to patients after the completion of FMT if pseudomembranes were present in a colonoscopy. This study indicates that in severe, and especially in severe/complicated CDI, multiple doses of FMT and anti-CDI antibiotics may be necessary to achieve treatment success.

A systematic review published in 2021 examined seven cost-utility studies comparing FMT to antibiotics alone or in combination with bezlotoxumab [80]. Delivery methods included colonoscopy, nasoduodenal or nasogastric infusion, enema, and oral capsules. FMT was deemed the most cost-effective in all studies. Five of the seven studies concluded that FMT was both cost saving and more effective than antibiotics. Though this review identified a number of potential methodological concerns and differences in study characteristics, the conclusion was still that FMT should be strongly considered in recurrent CDI as a cost-effective option compared to antibiotics alone.

Though long-term safety data is still needed for FMT, most documented short-term effects are not serious and include mostly GI effects, such as bloating and abdominal pain. There have been some reports of pathogens, such as multidrug resistant E. coli, SARS-CoV-2, and Mpox, being transmitted through FMT, which has led to the FDA issuing warnings and mandating testing for antibiotic-resistant bacteria [81,82]. As this treatment becomes more frequently utilized, the safety profile will likely become more clear.

FMT appears to be a beneficial therapeutic option for CDI, especially in patients with multiple recurrences. Logistics, such as donor stool acquisition, donor screening, formula variety, storage considerations, and personnel requirements, may limit the use of FMT. However, as this process becomes more standardized, some of those barriers may dissipate. More data is needed to determine if FMT might also be beneficial earlier in the treatment of CDI. 

The terms second or next-generation FMT have been used to describe microbiome-based products that aim to deliver the benefits of FMT while addressing some of the drawbacks. Recently, the ECOSPOR III trial was published with the aim to demonstrate the superiority of SER-109, an oral microbiome therapeutic, for reducing the risk of CDI recurrence up to 8 weeks after administration compared to a placebo [83]. This phase 3, double-blind, randomized, placebo-controlled trial included 182 patients with three or more episodes of CDI within 12 months. Patients underwent antibiotic therapy for CDI followed by magnesium citrate before receiving SER-109 or a placebo to avoid inactivation of the microbiome species. Of note, 99% of the participants were outpatients, and the majority underwent therapy with vancomycin. The primary endpoint was achieved, with 12% of SER-109 patients and 40% of placebo patients experiencing recurrence within 8 weeks (RR 0.32, 95% CI 0.18–0.58). These results remained statistically significant when patients were analyzed by age (greater than or less than 65 years) and by antibiotic (vancomycin or fidaxomicin). No serious adverse effects were noted, and the most common adverse effects were gastrointestinal in nature. 

RBX2660 is another microbiota-based live biotherapeutic containing several microorganisms prepared from screened donor stool [84]. This compound was recently approved by the FDA for the prevention of CDI recurrence in patients who have completed antibiotic therapy for CDI and is administered as a one-time rectal dose [85]. A phase 3 randomized, double-blind, placebo-controlled trial examined RBX2660 (*n* = 180) compared to a placebo (*n* = 87) in patients with at least one recurrence of CDI being treated with antibiotic therapy [82]. The primary endpoint was treatment success, which was defined as absence of CDI diarrhea within 8 weeks of the treatment. This was seen in 71% of the treatment group and 62% of the placebo group in the modified intention-to-treat (mITT) population. Using a Bayesian hierarchical model to analyze data from this trial and a previous phase 2 trial, the primary endpoint was estimated at 70% for the RBX2660 group and 58% for the placebo group in the mITT population, yielding a 12 percentage point treatment difference (95% CI 1.4–23.3). The posterior probability of superiority for RBX2660 compared to a placebo exceeded the lower threshold for demonstrating superiority. Adverse effects were seen in 56% of the treatment group and in 45% of the placebo group, with the majority of these being mild to moderate and gastrointestinal in nature. 

### 5.3. What Is Bezlotoxumab’s Role in Secondary Prophylaxis?

Bezlotoxumab is a human monoclonal antibody that binds *C. difficile* toxin B. It does not bind to toxin A [86]. A different monoclonal antibody (actoxumab) that is active against toxin A was evaluated in the MODIFY I trial and found not to improve clinical outcomes. It is administered as a single 10 mg/kg intravenous infusion over 60 minutes. A single dose of bezlotoxumab administered while receiving standard antibiotics for an initial or recurrent episode of CDI is sufficient, given its 19-day half-life. Bezlotoxumab should generally be avoided in patients with heart failure

A post-hoc analysis of the combined data for the MODIFY I and II trials showed that patients with zero risk factors for relapse did not benefit from receiving bezlotoxumab [87]. Similarly, patients <65 years of age with one or more risk factors did not derive a statistically significant benefit (25 vs. 32%). A univariate analysis of each risk factor (age 65 years or greater, history of CDI, immunocompromised, severe CDI, or RT 027/078/244) observed at least a 12.9% absolute reduction in CDI recurrence. RT 027/078/244 was the only subgroup where this reduction did not also yield a statistically significant difference. Bezlotoxumab had a favorable safety profile overall. However, patients with a history of CHF were more likely to have a serious adverse reaction of heart failure (13 vs. 5%, no *p*-value reported) or all-cause death (20 vs. 13%, no *p*-value reported) while receiving bezlotoxumab compared to a placebo [86]. 

Two retrospective cohorts were combined to evaluate fidaxomicin vs. a standard of care followed by bezlotoxumab [88]. The investigators found a slightly lower recurrence rate with bezlotoxumab (14% vs. 19%) that was not statistically significant with traditional multivariable or propensity score matched analyses. 

An early economic model suggested that bezlotoxumab was cost-effective compared to a placebo for subgroups of patients aged ≥65 years (ICER of USD 15298/QALY), immunocompromised patients (ICER of USD 12597/QALY), and patients with severe CDI (ICER of USD 21430/QALY) [89]. A more recent analysis comparing bezlotoxumab to other CDI therapies suggested that bezlotoxumab plus vancomycin was more cost-effective than vancomycin alone with an incremental net monetary benefit of USD 17,011 [58]. However, the analysis ultimately concluded that the addition of bezlotoxumab to vancomycin was dominated by standard fidaxomicin due to bezlotoxumab’s higher costs and lower QALY gained.

### 5.4. Is Vancomycin Effective as a Secondary Prophylaxis?

The aforementioned meta-analysis also evaluated the effectiveness of vancomycin as a secondary prophylaxis for CDI [22]. Seven retrospective cohort studies were included in the analysis. Four of the studies evaluated adults receiving antibiotics after an index CDI episode. The other three focused on immunocompromised patients (renal transplant, hematologic stem cell transplant, pediatric malignancies). The authors found a significant reduction in the risk of CDI when utilizing vancomycin (OR 0.31, 95% CI 0.13–0.71). The event rates were 16% (103/636) for vancomycin and 20% (232/1179) for the control group and varied widely between studies. The authors acknowledged this considerable heterogeneity being present in the meta-analysis (I2 71%, 95% CI 37–87%). This 4% absolute risk reduction is half of the benefit observed in studies evaluating vancomycin as a primary prophylaxis. While it appears that secondary prophylaxis with vancomycin is likely a beneficial intervention and can currently be implemented on a case-by-case basis, further research is needed to find the patient populations and/or risk factors to optimize its use.

## 6. Conclusions

This is an exciting time, given the expansion of treatment and prophylaxis options for CDI. Bezlotoxumab could represent a non-antimicrobial option for primary prophylaxis, but data are needed to confirm this approach. Further studies are also needed to help clinicians optimize patient selection, agent, and duration or primary prophylaxis. In the meantime, clinicians should use the risk factors examined to date (immunocompromised, elderly patients receiving antibiotics) for prescribing antimicrobial prophylaxis or probiotics. Metronidazole is still an option for patients with an initial mild CDI episode, but concerns about the impact of metronidazole resistance on outcomes remain. Most economic analyses show that fidaxomicin is the more cost-effective agent when taking all recurrences and healthcare costs into account. The challenge is working within healthcare systems where costs are fragmented and the benefit is not necessarily gained by the institution incurring the increased initial costs. The recent FDA approvals of commercially available FMT products will likely become part of the standard treatment for recurrent infections, given that costs are acceptable to payors. This approach would likely be preferred over secondary prophylaxis with antimicrobials, given their relative impacts on the microbiome. Bezlotoxumab is a viable alternative for patients without congestive heart failure. Head-to-head comparisons of these options for secondary prophylaxis will also help further inform these decisions.

## Figures and Tables

**Table 1 microorganisms-11-00387-t001:** Treatment Strategies for CDI.

	IDEA/SHEA	ACG	ESCMID	Current Review
**Preferred Regimens for an Initial CDI Episode**				
Non-severe	Fidaxomicin	Fidaxomicin or vancomycin (metronidazole for low-risk only)	Fidaxomicin	Fidaxomicin or vancomycin (metronidazole for low-risk only)
Severe	Fidaxomicin	Fidaxomicin or vancomycin	Fidaxomicin or vancomycin	Fidaxomicin or vancomycin
Fulminant/complicated	High-dose vancomycin + IV metronidazole	High-dose vancomycin ± IV metronidazole	Vancomycin or fidaxomicin	
**Preferred Regimens for Recurrent CDI Episodes**				
First recurrence	Fidaxomicin	Fidaxomicin or tapered/pulsed vancomycin	First-line: Fidaxomicin or the addition of bezlotoxumab (tailored based on treatment regimen for the initial episode)	More data needed. Lowest recurrence rates with fidaxomicin
Second recurrence	Fidaxomicin, vancomycin tapered and pulsed regimen, vancomycin followed by rifaximin, FMT	Not specifically addressed	FMT or standard regimens and bezlotoxumab, if not used previously (tailored based on past treatment regimens)	

**Table 2 microorganisms-11-00387-t002:** Preventative Strategies in CDI.

	IDEA/SHEA	ACG	ESCMID	Current Review
**Primary Prophylaxis to Prevent an Initial CDI Episode**				
Probiotics for primary prevention	Insufficient evidence	Recommends against	Not routinely recommended	Optimal role to be defined in populations with >5% risk of CDI
Antimicrobial prophylaxis	Not specifically addressed	Not specifically addressed	Not routinely recommended	To be considered in patients with sufficiently high baseline risk
PPI Discontinuation	Insufficient evidence to recommend discontinuation as a prevention measure	Recommends against discontinuation if an appropriate indication exists	Use should be reviewed	Ensure PPIs have a valid indication Used cautiously in high-risk patients
**Strategies to Prevent Recurrent CDI episodes**				
Antimicrobial prophylaxis	Insufficient evidence to recommend suppressive or prophylactic agents	Suppressive vancomycin may be used in patients who cannot undergo or fail FMT and require frequent antibiotics; vancomycin prophylaxis may be considered during antibiotic use in patients with CDI history who are at high risk of recurrence	Prophylactic therapy may be warranted in select patients with multiple recurrences	Considered on a case-by-case basis
FMT	≥2 recurrences	≥2 recurrences	≥2 recurrences	Current recommendation is ≥2 recurrences. Role in primary CDI is of future interest
Bezlotoxumab	Recurrent infection in the last 6 months	Considered in patients at high risk for recurrence	First and subsequent recurrences	Main advantage is that it can be administered during antibiotic therapy. Its comparative effectiveness to FMT is unknown

## Data Availability

All data are stored on the VA Informatics and Computing Infrastructure servers. These data are not publicly available due to the ability to identify patients through health records. Data available on request due to privacy restrictions.

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
