# Peer review of "Controversies in the Prevention and Treatment of Clostridioides difficile Infection in Adults: A Narrative Review"

_microorganisms, 2023, doi:10.3390/microorganisms11020387_

Round 1
Reviewer 1 Report
In this narrative review, the authors have reviewed available literature regarding the prevention or treatment of CDI and have focused on disagreements between the IDSA/SHEA and ACG guidelines as well as articles that have been published since the updates.
The methodology is correct. The manuscript has a good sound.
Author Response
Thank you for your favorable review and your valuable time.

Reviewer 2 Report
1. The US is just one country. Please also refer to European guidelines - European Center for Disease Prevention and Control (ECDC), European Society for Paediatric Gastroenterology, Hepatology and Nutrition ESPGHAN and WHO regarding epidemiological data, prophylactics and treatment recommendations.
2. Please use current taxonomic nomenclature for species previously named Lactobacillus, e.g. Lactobacillus casei is now Lacticaseibacillus casei.
3. Please describe the CDI diagnostic algorithms recommended in the US and EU.
4. Line 95: B. Bifidum - convert to B. bifidum.
5. Latin names of bacterial taxa must be written in italics
Author Response
Reviewer #2
- The US is just one country. Please also refer to European guidelines - European Center for Disease Prevention and Control (ECDC), European Society for Paediatric Gastroenterology, Hepatology and Nutrition ESPGHAN and WHO regarding epidemiological data, prophylactics and treatment recommendations.
Thank you for this thoughtful critique. We have now incorporated the European guidelines and WHO statistics into our review (Table 1). We feel that addressing pediatric data or guidelines is outside of the scope of this review. We have now specified in the title, abstract and introduction that our review is specifically focused on adult patients.
- Please use current taxonomic nomenclature for species previously named Lactobacillus, e.g. Lactobacillus caseiis now Lacticaseibacillus casei.
We have corrected this throughout the paper. Thank you for pointing this out.
- Please describe the CDI diagnostic algorithms recommended in the US and EU.
We are very appreciative of this comment and feel that it is very important. That being said, we do feel that this topic is outside of the scope of this extensive article regarding prevention and treatment of CDI. In fact, one could consider writing an entire review solely on the diagnostic approaches used for CDI.
- Line 95: B. Bifidum - convert to B. bifidum.
Thank you, we have corrected throughout.
- Latin names of bacterial taxa must be written initalics
Thank you, we have corrected throughout.

Reviewer 3 Report
Very interesting and exaustive narrative review on the treatment and prophylaxis options by Clostridioides difficile infection (CDI). Only one request: at lane 95 B. Bifidum should become B. bifidum. Great job.
Author Response

(The authors gave the same response as above.)

Round 2
Reviewer 2 Report
I accept the manuscript
Author Response
Thank you for your time and consideration of our manuscript.